## Classics

*Antirrhinum*; *Arabidopsis*; evo-devo; floral homeotics; floral development; MADS box.

**Corresponding author:**
Kay Schneitz;
Email: kay.schneitz@tum.de

# The 1991 review by Coen and Meyerowitz on the war of the whorls and the ABC model of floral organ identity

Kay Schneitz

Plant Developmental Biology, TUM School of Life Sciences, Technical University of Munich, Munich, Germany

### Abstract

The 1991 review paper by Coen and Meyerowitz on the control of floral organ development set out the evidence available at that time, which led to the now famous ABC model of floral organ identity control. The authors summarised the genetic and molecular analyses that had been carried out in a relatively short time by several laboratories, mainly in *Arabidopsis thaliana* and *Antirrhinum majus*. The work was a successful example of how systematic genetic and molecular analysis can decipher the mechanism that controls a developmental process in plants. The ABC model is a combinatorial model in which each floral whorl acquires its identity through a unique combination of floral homeotic gene activities. The review also highlights the similarities in the regulation of floral organ identity between evolutionarily distant plant species, emphasising the general relevance of the model and paving the way for comprehensive studies of the evolution of floral diversity.

The late 1980s and early 1990s were an exciting time in plant developmental biology. Within the period of about a year, from November 1990 to September 1991, two reviews on the role of homeotic genes in floral development appeared in Science (Schwarz-Sommer et al., 1990) and Nature (Coen & Meyerowitz, 1991). The two reviews summarised the pioneering work that had been performed mainly in the laboratories of Enrico Coen at the John Innes Centre, Norwich, UK, Elliot M. Meyerowitz at the California Institute of Technology, Pasadena, CA, USA, and by Zsuzsanna Schwarz-Sommer and Hans Sommer from Heinz Saedler's department at the Max Planck Institute for Plant Breeding Research, Cologne, Germany. In October 1991, yet another landmark paper was published in Nature, describing the exciting work of the laboratory of Gerd Jürgens, then at the Ludwig-Maximilians-University, Munich, Germany, later at the University of Tübingen, Tübingen, Germany, which successfully performed a systematic genetic analysis of the body organisation of the *Arabidopsis* embryo (Mayer et al., 1991).

What made these papers special? Two groundbreaking genetic analyses had identified the key genes that regulate segmental identity and segmentation in *Drosophila* (Lewis, 1978; Nüsslein-Volhard & Wieschaus, 1980). They paved the way for a series of subsequent molecular and genetic studies that identified a regulatory network of transcription factors and signalling components that control these and other important developmental decisions (Gehring, 1993; Morata & Lawrence, 2022). The question was whether such a genetic and molecular strategy could be successful in plants. It should be remembered that plant developmental biology was not a new field of research at that time, but it was certainly an underexplored one. In fact, very little was known about the genetic and molecular mechanisms that regulate plant development (Steeves & Sussex, 1989). There was a general belief that plant development, because of its inherently more flexible nature, must be controlled by mechanisms quite different from those that govern animal development. Unimaginable from a present-day perspective, a considerable number of plant scientists even believed that genes did not play a significant role in plant development. For many developmental biologists, including myself, who, like other aspiring plant developmental biologists of my generation, had an animal background (Schneitz et al., 1993), the work summarised in these papers embodied the certainty that a coherent genetic and molecular approach was feasible and could lead to fundamental insights into the mechanisms underlying developmental processes in plants.

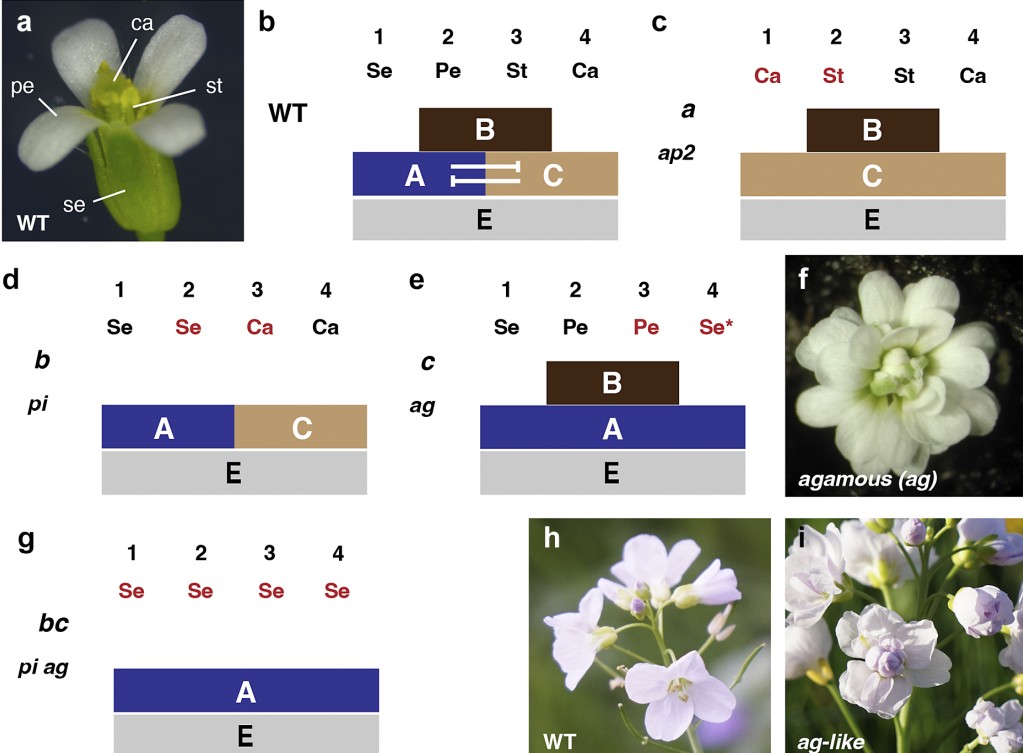

**Figure 1.** The ABC model and the control of floral organ identity. (a) Mature wild-type flower of *Arabidopsis thaliana*. (b) Schematic representation of the ABC model. The four whorls and the corresponding floral organs are indicated as well as the A, B and C regions. The arrows denote that A and C functions act antagonistically. The spatial extent of the E function is also displayed. (c) Representation of the *Arabidopsis apetala2* (*ap2*) mutant phenotype (defective in A function) and the explanation based on the ABC model. (d) The *Arabidopsis pistillata* (*pi*) mutant phenotype (loss of B function). (e) The *Arabidopsis agamous* (*ag*) mutant phenotype (defective in C function). The Se∗ notation indicates the defect in floral meristem termination as shown in (f). (f) Top view of a mature flower of the *Arabidopsis ag* mutant. Note the abundance of petals. The *ag* mutant is also defective in floral meristem termination and thus produces a flower within a flower. (g) Floral organisation of an *Arabidopsis* mutant lacking *PI* and *AG* activity (defective in B and C functions). (h) A mature flower of wild-type *Cardamine pratensis*. (i) An *ag*-like flower of a natural variant of *Cardamine pratensis*. Compare with (f). Abbreviations: ca, carpel; pe, petal; se, sepal; st, stamen. Images in (h,i) courtesy of Thomas Huber.

Obviously, the two reviews on floral homeotic genes did not come out of nowhere. In fact, floral mutants had been studied for centuries (Meyerowitz et al., 1989). However, they provided a concise summary of the painstaking genetic work on floral homeotic mutants in *Arabidopsis* (Bowman et al., 1989; 1991; Komaki et al., 1988; Kunst et al., 1989; Meyerowitz et al., 1991; Schultz & Haughn, 1991) and *Antirrhinum* (Carpenter & Coen, 1990; Coen, 1991; Coen et al., 1991; Stubbe, 1966). The two reviews also highlighted the initial molecular identification and characterisation of the floral homeotic genes *deficiens* (*def*) (Sommer et al., 1990) and *floricaula* (*flo*) (Coen et al., 1990) in *Antirrhinum* and *AGAMOUS* (*AG*) (Yanofsky et al., 1990) in *Arabidopsis*. Here, I highlight the 1991 review by Coen and Meyerowitz titled 'The war of the whorls: genetic interactions controlling flower development'. The reason is that in this review, results from the analysis of floral mutants from two evolutionary divergent species, *Antirrhinum majus* and *Arabidopsis thaliana*, were combined to propose a general model for the regulation of floral organ identity. The earlier review by Schwarz-Sommer et al. focussed on *Antirrhinum* floral development. The Coen and Meyerowitz review also discusses other aspects of floral development, including the determination of floral meristem identity, the control of floral organ number and floral symmetry. Here, I focus on floral organ identity because, from my perspective, this is what it is best known for.

To understand how floral homeotic genes regulate floral organ identity, it is necessary to first look at the bauplan of the flowers of *Arabidopsis* and *Antirrhinum* (Fig. 1a). They consist of four concentric units called whorls. Each whorl is distinct and characterised by a unique type of floral organ. The outermost whorl 1 contains sepals; the next inner whorl 2 bears petals; whorl 3 features stamen and the innermost whorl 4 contains carpels, which carry the ovules. The two outermost whorls bearing the sepals and petals form the perianth. Organ number per whorls varies between whorls and the two species. For example, in *Arabidopsis*, whorl 1 bears four sepals, whereas whorl 3 contains six stamens. In *Antirrhinum*, whorl 1 bears five sepals and whorl 3 ultimately contains four stamens. Pattern formation in the flower thus leads to the formation of repeating developmental units, the concentric whorls, each of which is endowed with its own particular identity, as evidenced by the different types of floral organs of varying numbers that they form.

Homeotic genes are characterised by their respective mutant phenotypes. A defect in a homeotic gene disrupts the specification of early progenitor cells and eventually leads to the substitution of one organ type for another (Bateson, 1894). Careful systematic analysis of the type of organ transformation and where it occurs in the mutant flower led to the realisation that floral homeotic genes do not affect individual organs but three distinct and overlapping regions, each spanning two neighbouring whorls (Fig. 1b–g). The regions were named A, B and C, elaborating on a notation proposed in an earlier review by George Haughn and Chris Somerville (Haughn & Somerville, 1988). Region A spans whorls 1 and 2, region B whorls 2 and 3 and region C whorls 3 and 4. For example, defects in the flowers of the *Arabidopsis* floral homeotic mutant

*apetala 2* (*ap2*) are restricted to region A as they form carpels rather than sepals in whorl 1 and stamens rather than petals in whorl 2 (Fig. 1c). A similar phenotype can be observed for *ovulata* (*ovu*) mutants in *Antirrhinum*. However, *ovu* mutants carry gain-of-function alleles of *PLENA* (*PLE*) (Bradley et al., 1993) (see below for problems with the A function). The *pistillata* (*pi*) mutant of *Arabidopsis* and the *deficiens* (*def*) mutant of *Antirrhinum* are affected in region B. Flowers of *pi/def* mutants carry sepals and carpels instead of petals and stamens in whorls 2 and 3, respectively (Fig. 1d). Plants with loss-of-function defects in the *Arabidopsis* gene *AGAMOUS* (*AG*) or its *Antirrhinum* homologue *PLE* bear flowers with defects restricted to region C with stamens in whorl 3 being substituted by petals and carpels in whorl 4 being replaced by sepals or variable structures (Fig. 1e,f). In addition, multiple homeotic genes can contribute to whorl identity. This is obvious in *Antirrhinum* where, for example, the combined action of *DEF* and *GLOBOSA* (*GLO*) regulate the identity of whorls 2 and 3 (region B). In *Arabidopsis,* a similar observation was made for *PI* and *APETALA3* (*AP3*).

Analysis of *Arabidopsis* and *Antirrhinum* floral homeotic mutants eventually led to the now famous ABC model of floral organ identity control that was outlined so lucidly in the review by Coen and Meyerowitz. At its core, it is a combinatorial model. Multiple homeotic floral genes are assumed to operate in the three overlapping A, B and C regions providing each whorl with a unique combination of A, B and C regulatory functions (originally named a , b and c in Coen and Meyerowitz's review) (Fig. 1b). The identity of whorl 1 sepals is based on the A function, of whorl 2 petals on a combination of A and B functions, of whorl 3 stamens on a combination of B and C functions and of whorl 4 carpels on C function. The model further states that the B function domain does not depend on either the A or C function genes. Finally, it includes an antagonistic interaction between A and C functions which results in the absence of C function activity in whorls 1 and 2 and A function activity in whorls 3 and 4.

What happens if there is no floral homeotic activity at all? *Arabidopsis* plants impaired in A, B and C functions, as in *ap2 ap3 ag* triple mutants, form flower-like structures made entirely of leaf-like organs (Bowman et al., 1991). This finding suggests a leaf-like 'ground state' that is modified by the activity of homeotic genes. The idea that floral organs derive from a leaf-like ground state is reminiscent of surprisingly similar ideas put forward by two eminent German scholars of the eighteenth century. The embryologist Caspar Friedrich Wolff set out his hypothesis in his 'Theoria Generationis' (Wolff, 1759) and the poet Johann Wolfgang von Goethe formulated his thoughts in his 'Metamorphose der Pflanzen' (Goethe, 1790). Interestingly, both scholars derived their ideas from comparisons between regular and irregular flowers of a species, where, for example, stamens were transformed into petals. Such examples can be observed in nature today (Fig. 1h,i). In a sense, the two authors pioneered the use of a genetic analysis to study developmental processes, where one learns about the regular function of a gene by studying the consequences of the absence of its function, centuries before such an approach was commonly accepted and successfully applied to the study of development.

It is important to note that this elegant ABC model was derived entirely from genetics. It provided a robust framework that repeatedly proved itself in genetic experiments. The key point is that a unique combination of the functions A, B and C provides a pre-pattern in the floral meristem that ultimately determines the identity of each whorl. It conveniently explained all single and multiple mutant phenotypes. However, it did not provide ready insight into the molecular mechanism regulating floral organ identity. Nevertheless, it made testable predictions. For example, it proposed that activity of A, B and C functions is restricted to the respective A, B and C regions. Molecular analysis of the structure of floral homeotic genes and their mode of action revealed that in most cases the spatial regulation of gene activities underlying A, B and C functions occurs at the RNA level in young floral meristems. For example, in situ hybridisation data suggested that expression of *DEF* or *AG* is restricted to the B and C regions, respectively (Schwarz-Sommer et al., 1990;Sommer et al., 1990; Yanofsky et al., 1990). In addition, expression analysis further revealed that the A function gene *APETALA2* (*AP2*) inhibits the expression of the C function gene *AG* in future whorls 1 and 2 (Drews et al., 1991).

In *Drosophila,* the homeotic genes regulating segment identity encode homeobox transcription factors (Gehring, 1993). Interestingly, floral homeotic genes also encode transcription factors but not of the homeobox family. Most of them, such as *DEF* or *AG*, encode transcription factors of the MADS-box class (Sommer et al., 1990; Yanofsky et al., 1990), named after the conserved DNA-binding motif shared by the canonical members of this gene family, which include yeast *MCM1*, *AG*, *DEF* and human *SRF* (Schwarz-Sommer et al., 1990). Thus, there is an interesting parallel logic in the regulation of regional identity between animals and plants (Meyerowitz, 1997). In both instances, the overlapping spatial expression patterns of transcription factor genes determine the identities of repetitive body regions, segments in *Drosophila* and floral whorls in plants.

The elegant original ABC model was immediately widely accepted. However, it soon became apparent that the A, B and C genes were not sufficient for floral organ identity (Krizek & Meyerowitz, 1996; Mizukami & Ma, 1992), indicating that additional components were missing. The missing factors turned out to be the four closely related and redundantly acting *SEPALLATA* (*SEP*) genes, also members of the MADS-box gene family (Ditta et al., 2004; Pelaz et al., 2000). The *SEP* genes were assigned the E function required for petal, stamen and carpel identity, and thus the modern standard model is known as the ABCDE model [the D function is required for ovule development (Angenent et al., 1995; Colombo et al., 1995) which happens within the carpel and for simplicity is not discussed here]. The genetic and molecular evidence led to the notion that the combinatorial property of the original ABC model relies on region-specific multimeric complexes of MADS-box transcription factors of the A, B and C classes in combination with one of the SEP factors and that these higher-order complexes are required and sufficient for promoting floral organ identity (Honma & Goto, 2001; Pelaz et al., 2001). To reflect this molecular scenario, the 'floral quartet model' of floral organ identity has been proposed, which emphasises the different multimeric protein complexes (Theissen, 2001). Taken together, the ABCE/floral quartet model provides a convenient scenario for the regulation of floral organ identity in *Arabidopsis*. But is it universally applicable?

Through the comparisons between the regulators of floral development of *Arabidopsis* and *Antirrhinum*, the 1991 review by Coen and Meyerowitz also immediately demonstrated the importance of a comparative evolutionary developmental (evo-devo) genetics approach in assessing the general relevance of the findings. Their discussion of the topic highlighted that despite their taxonomic distance *Antirrhinum* and *Arabidopsis* share extensive homology with respect to the genetic and molecular mechanisms regulating floral organ identity. The ABC model inspired numerous

laboratories to embark on a fruitful scientific journey to investigate the genetic and molecular basis of the evolution of floral organ identity (Chanderbali et al., 2016; Kramer, 2019; Theißen et al., 2016).

The evolutionary studies revealed a high degree of conservation but also interesting differences in the regulation of floral organ identity between plant species. For example, there is consensus in the literature that B and C function genes are required for reproductive organ identity across the angiosperms (Di Stilio, 2011). In addition, B and C class genes were found to be present in gymnosperms, where they are also expressed in reproductive organs (Gramzow et al., 2014; Melzer et al., 2010; Winter et al., 1999). The results suggest functional conservation between B/C class genes that seems to predate the emergence of angiosperms. However, the A function has been contentious from the beginning. For example, in contrast to *Arabidopsis*, no recessive loss-of-function alleles of A-function genes affecting the entire perianth (whorls 1 and 2) were identified in *Antirrhinum* or other investigated plants (Litt, 2007). There is additional evidence that is difficult to reconcile with the original ABC model and often relates to the proposed dual role of the A function in controlling organ identity in whorls 1 and 2 and repressing C function in this region. Indeed, it is debated whether there is an A function equivalent to that proposed for *Arabidopsis* outside the Brassicaceae, or whether there is indeed an A function in any species (Causier et al., 2010). To address this central problem, Causier et al. proposed to replace the classical A function with a new (A) function resulting in an (A)BC model. The (A) function is flexible and expandable and can encapsulate multiple regulatory cascades. It first provides floral context by initially controlling floral meristem identity and subsequently regulates the B and C functions.

Another example relates to representatives of early-diverging ('basal') angiosperm lineages, including Amborella, Nymphaeales, magnoliids and basal eudicots. Corresponding representatives show an impressive floral diversity that encompasses more gradual transitions in organ identity, often including an undifferentiated perianth carrying organs called tepals (Chanderbali et al., 2016). This is in contrast to the perianth organisation in *Arabidopsis thaliana* and *Antirrhinum majus*, which are highly derived species within the rosids and asterids, respectively, and bear distinct sepals and petals. The gradual transition in floral morphology during the early evolution of flowers was associated with broader expression domains of floral identity regulator genes ['shifting border' (Bowman, 1997) or 'sliding boundary' (Kramer et al., 2003) models] and coupled with functional gradients such that there is decreasing expression/functional influence towards the edges of each expression domain ('fading borders' model) (Buzgo et al., 2004; Chanderbali et al., 2010). In this evolutionary model, the early 'fading boundaries' system with broadly overlapping expression domains evolved into the ABCE/(A)BC framework, including the sharply delineated expression domains of class A, B and C genes.

In retrospect, the impact of Coen and Meyerowitz's (1991) review on the regulation of floral organ identity was profound. Although the original simple model has been modified and remains under scrutiny today, variations of the ABC model continue to form the basis of our understanding of how floral organ identity is regulated at the genetic and molecular level (Ali et al., 2019; Bowman et al., 2012; Causier et al., 2010; Chanderbali et al., 2016; Kramer, 2019; Rijpkema et al., 2010; Rümpler & Theißen, 2019; Theißen et al., 2016). The authors not only summarised the exciting new insights of the period culminating in the original ABC model, but convincingly highlighted the value of systematic molecular and genetic analysis in unravelling the regulation of plant development. Furthermore, the review illustrated the power of an evo-devo genetics approach and helped pave the way for rigorous molecular analysis of the evolutionary basis of the dazzling floral diversity that surrounds us today.

## Acknowledgements

I thank members of my lab for fruitful discussions and the three referees for helpful comments.

**Financial support.** Work in my lab is supported by the Deutsche Forschungsgemeinschaft (Grant Nos. SFB924/TPA2, FOR2581/TP7 and SCHN 723/11-1).

**Competing interest.** The author declares no competing interest.

**Authorship contribution.** K.S. conceived and wrote this review.

**Data availability statement.** Data availability is not applicable to this article as no new data were created or analysed in this study.

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
