## [Reviewer Report]

Dear editor,

please find enclosed my invited review for a paper of the “classics” section. I had no example to guide me and thus I hope it fits and corresponds to the expectations.

I added a figure to the main text but could not come up with a separate graphical abstract that does not replicate the figure. I simply uploaded a section of this figure as GA. Otherwise I could not have finished the submission. I don’t think it is suitable and if someone at the journal has a good idea please go ahead.

Sincerely, Kay Schneitz

---

## [Reviewer Report]

Dear Pr Schneitz,

We thank you very much for your appreciated manuscript highlighting the importance of the pioneer work by Coen and Meyerowitz.

Your manuscript has been now revised by three reviewers (please find their comments below) with interesting comments to improve the manuscript. In agreement with them, I think it would be interesting to develop a little bit the limitations of the ABC model and the main alternative models proposed nowadays, and to synthetize the result of your “AI experiment” at the end of the manuscript, and develop your thoughts/conclusions on it: what are the questions triggered?

We would be happy to receive a corrected version of your manuscript when it is ready.

We thank you again for having submitted your manuscript to Quantitative Plant Biology.

Thank you very much in advance,

Looking forward to reading you

Best regards

Daphné Autran

---

## [Reviewer Report]

Dear Kay,

We thank you very much for all the revisions in the very nice new version of your manuscript. This new version was submitted to the reviewers whose response are very positive (please find copy of their comments below), yet reviewers 1 and 3 suggest a few minor revisions on the newly added sections. I suggest to include these minor revisions, except for the revisions of Figure 1 suggested by Reviewer 1, which seem to me optionnal, since the current figure is clear and the wild ag mutant is represented. However, you might choose to change it according to the reveiwer suggestions.

Many thanks again for your contribution to Quantitative Plant Biology and for all your work.

Best regards

Daphné

Reviewer 1:

The author has addressed most of my concerns, except point 3) where perhaps they feel necessary to keep the full explanation of all mutant classes (Lines 114-124, see related comment on Fig 1, where I suggest keeping only WT and ag diagrams).

Overall, I thought the original figure was more effective. I had not requested for it to change, but rather to provide more comment on the mutant in the wild.

Other minor points that arise from newly added sections are listed below.

Detailed comments (by line number):

Line 193: Please provide a reference for D function

Lines 218-222: citations are needed here also (B and C conservation across angiosperms, reviewed e.g. in 10.1002/bies.201100040; presence in gymnosperms)

238: Incorrect use of term “ancestral”, no extant organisms are ancestral, they may exhibit ancestral character states, or they may be representative of early-diverging lineages. I suggest instead “Another example relates to representatives of ancestral angiosperm lineages…” OR “Another example relates to early-diverging (“basal”) angiosperm lineage representatives…” Deleting that phrase will also take care of the issue that Amborella and Nymphaeles are basal angiosperms (ANITA grade members), while magnoliids are another lineage, and neither are eudicots. The Eudicot lineage is more recent and contains Core Eudicots (Arabidopsis, Antirrhinum, etc.) and non-core or early-diverging Eudicots (Ranunculids et al). For examples on abc model in early diverging Eudicots the author may refer to work done in the Ranunculids Aquilegia (cited), Thalictrum and California poppy.

241 …more gradual transitions in organ identity

253: the incorporation of the fading borders models is a nice addition, the author could consider adding that ‘shifting border’ (Bowman, 1997) and ‘sliding boundary’ (Kramer et al., 2003) model variations were also proposed to explain the diversity of flower morphology beyond basal angiosperms, such as the tepals of monocot lilies with perianth consisting of two whorls of equally petaloid organs expressing B genes.

259 by “variants” maybe the author means variations? It could be mistaken with mutants.

Figure 1:

The addition of the E class is an improvement, but the block should not go beyond the A+C blocks in my opinion, as this would suggest expression/function somewhere beyond whorls 1 and 4.

I personally preferred the previous version of this figure, with all photographs arranged together on one side, rather than interspersed with graphs, and with the original example of the agamous mutant in a wild population, I thought that was an especially unique contribution. Other examples of the use of abc mutants in the wild can be cited when referring to that panel, e.g. 10.1093/jxb/erl158; 10.1093/aob/mcad069 and 10.1016/j.cub.2022.01.066.

WT and c (ag) mutant models would be sufficient in this figure, as those are shown in the photo panels and the rest is repetitive and has been depicted too many times to warrant another reproduction.

Reviewer 2:

The author has addressed all my comments: he has added one paragraph discussing the E-function and the quartet model, one discussing the limitations of the A function, and another one about the applicability of the ABC model to early-diverging angiosperms. I think the current manuscript is a nice addition to the existing litterature on the topic.

Reviewer 3:

The updated version of this review is a significant improvement on the original. While I quite liked the ChatGPT section of the original, the sections that now replace it, as suggested by the other reviewers, are much more important additions to this article.

I have some minor points:

On page 7, lines 113-114 you say that the Antirrhinum ovu mutant resembles Arabidopsis ap2, leaving the reader to infer that OVU is an Antirrhinum A-function gene. Later, on page 11, lines 224-227, you correctly say that no null A-function mutants with homeotic changes to both whorls 1 and 2 have been identified in Antirrhinum or other species. This is slightly at odds with the earlier description of the ovu mutant. Somewhere it needs to be made clear that ovu is a PLE gain-of-function allele (Bradley et al., 1993) and not a true A-function mutant.

Line 170: please change “a function” to “A function”

---

## [Reviewer Report]

Dear Daphne and Olivier,

I followed Daphne’s suggestions for this final revision (R2). I kept figure 1 as in R1. I hope the MS is now acceptable. Cheers, Kay

---

## [Reviewer Report]

Dear Kay,

I sincerely apologize for the waste of time with your last version due to my mistakes with the system.

Many thanks for your revised final version which includes all the last reviewers minor corrections.

Thanks again for your contribution to Quantitative Plant Biology

Best regards

Daphné